# Polyacetylene Isomers Isolated from *Bidens pilosa* L. Suppress the Metastasis of Gastric Cancer Cells by Inhibiting Wnt/*β*-Catenin and Hippo/YAP Signaling Pathways

**DOI:** 10.3390/molecules28041837

**Published:** 2023-02-15

**Authors:** Jing Cai, Song-Yun Shi, Fan Cheng, Min Wei, Kun Zou, Xiao-Qin Yu, Jian-Feng Chen

**Affiliations:** Hubei Key Laboratory of Natural Products Research and Development, College of Biological and Pharmaceutical Sciences, China Three Gorges University, Yichang 443002, China

**Keywords:** polyacetylene isomers, migration, invasion, Wnt/*β*-catenin, Hippo/YAP

## Abstract

(*E*)-7-Phenyl-2-hepten-4,6-diyn-1-ol (**1**) and (*Z*)-7-Phenyl-2-hepten-4,6-diyn-1-ol (**2**) are isomeric natural polyacetylenes isolated from the Chinese medicinal plant *Bidens pilosa* L. This study first revealed the excellent anti-metastasis potential of these two polyacetylenes on human gastric cancer HGC-27 cells and the distinctive molecular mechanisms underlying their activities. Polyacetylenes **1** and **2** significantly inhibited the migration, invasion, and adhesion of HGC-27 cells at their non-toxic concentrations in a dose-dependent manner. The results of a further mechanism investigation showed that polyacetylene **1** inhibited the expressions of Vimentin, Snail, *β*-catenin, GSK3*β*, MST1, YAP, YAP/TAZ, and their phosphorylation, and upregulated the expression of E-cadherin and p-LATS1. In addition, the expressions of various downstream metastasis-related proteins, such as MMP2/7/9/14, c-Myc, ICAM-1, VCAM-1, MAPK, p-MAPK, Sox2, Cox2, and Cyr61, were also suppressed in a dose-dependent manner. These findings suggested that polyacetylene **1** exhibited its anti-metastasis activities on HGC-27 cells through the reversal of the EMT process and the suppression of the Wnt/*β*-catenin and Hippo/YAP signaling pathways.

## 1. Introduction

Gastric cancer (GC) is the third leading cause of cancer-associated mortality in the world [1]. With the development of new treatment strategies, the long-term survival outcome of patients with GC has significantly improved, especially for pre-metastatic early patients, where the 5-year survival rate can exceed 95% [2]. Unfortunately, due to the lack of early clinical signs, GC is difficult to observe in the beginning and about 80% of patients have advanced-stage GC at the initial diagnosis; this diagnosis is usually accompanied by lymph node or even distant organ metastasis, which leads to the loss of the best surgical window and a serious restriction of the prognosis and the survival of patients [3]. Therefore, it is critical to find safe and effective candidate drugs that inhibit the metastasis of GC cells.

Tumor metastasis is a complex, multifactorial dynamic process that mainly depends on the potential of tumor cell migration and invasion [4]. The canonical Wnt/*β*-catenin signaling pathway is mostly considered to be closely related to tumor cell migration and invasion [5]. The activation of the Wnt/*β*-catenin signaling pathway can promote tumor cell adhesion by forming a cadherin–catenin complex, enhancing the epithelial–mesenchymal transition (EMT) process, promoting clone initiation and invasion, and thus, aggravating tumor metastasis [6]. In addition, the Hippo/YAP signaling pathway was also revealed to be intricately related to tumor invasion and metastasis in recent years, especially in gastric cancer [7,8,9]. The expression and nuclear localization proportion of YAP/TAZ, which is the core protein of the Hippo/YAP signaling pathway, were both abnormally elevated in GC cells, indicating the special interlinkage between this pathway and the physiological and pathological processes of GC metastasis [10]. Interestingly, these two pathways are both activated by promoting the translocation of their core signal molecules (*β*-catenin and YAP/TAZ) to the nucleus, binding with transcription factors, and then transcribing downstream target genes [11]. Furthermore, frequent cross-talk occurs between Wnt/*β*-catenin and Hippo/YAP signaling pathways. Phosphorylation YAP/TAZ can integrate with *β*-catenin, causing *β*-catenin to be retained in the cytoplasm, thus inhibiting the transcriptional activity of *β*-catenin [12,13]. YAP can induce the inactivation of GSK3*β*, which, in turn, stabilizes cytosolic *β*-catenin and promoted its nuclear translocation of *β*-catenin [14]. Given this, some scholars pointed out that the creation of innovative anti-metastasis medications based on the intracellular microenvironment of Wnt/*β*-catenin and Hippo/YAP signaling pathways may be the key to producing promising results in the field of tumor control.

(*E*)-7-Phenyl-2-hepten-4,6-diyn-1-ol (polyacetylene **1**) and (*Z*)-7-Phenyl-2-hepten-4,6-diyn-1-ol (polyacetylene **2**) (Figure 1) are a pair of natural polyacetylene isomers isolated from the Chinese traditional herb *Bidens pilosa* L. by our research group. These two compounds are characterized by the presence of a conjugated system of two carbon–carbon triple bonds and a double bond. Extensive natural polyacetylenes were isolated from different plants, fungi, and marine organisms in recent decades and extensive attention was paid owing to their unique chemical structures and diverse bioactivities, including tumor suppression, immunity regulation, depression resistance, and neural protection [15]. For example, phenyl-1,3,5-heptatriyne showed significant cytotoxicity against human hepatoma HepG2 cells and human colon cancer CaCO-2 cells with IC_50_ values of 0.49 and 0.7 µg/mL, respectively [16]. In addition, four falcarindiol-type polyacetylenes isolated from *E. tricuspidatum* exhibited significant inhibition of cell proliferation on human malignant melanoma SK-MEL-28 cells, human pulmonary carcinoma A549 cells, and human breast cancer MCF-7 cells with IC_50_ values ranging from 0.3 to 29 μM [17]. It is worth mentioning that these polyacetylenes with anti-tumor activity in vitro shared a conjugated system of two carbon–carbon triple bonds and a double bond. However, there is no report on the inhibitory activity and possible mechanism of polyacetylene compounds on tumor cell migration and invasion.

Therefore, this study focused on the influence of two polyacetylene isomers on tumor metastasis, as well as the Wnt/*β*-catenin and Hippo/YAP signaling pathways to deeply understand the molecular mechanisms. Overall, our research provided new insight for developing novel, secure, and effective anti-GC metastasis drugs from natural polyacetylenes compounds.

## 2. Results

### 2.1. Effects of Polyacetylenes ***1*** and ***2*** on Cell Viability

As shown in Table 1, polyacetylene **1** displayed relatively obvious proliferation inhibitory activity against undifferentiated gastric cancer HGC-27 cells with an IC_50_ value of 52.83 μM, followed by human breast cancer MDA-MB-231 cells with an IC_50_ value of 73.92 μM. While other tested cells were all insensitive to polyacetylene **1** (IC_50_ values more than 100 μM). In addition, polyacetylene **2** was inactive in all tested cells (IC_50_ value more than 200 μM), including HepG2 cells, NCI-N87 cells, GES-1 cells, and MDCK cells. Based on the above results, the HGC-27 cells were used for further investigation in the anti-migration and invasion activities of polyacetylenes **1** and **2**.

### 2.2. Determination of Non-Cytotoxic Concentrations

Conventional drugs that inhibit the migration and invasion of cells are often achieved by restraining proliferation, which cannot purely reflect the anti-migration and invasion activity of drugs [18,19]. Therefore, this study evaluated the effects of polyacetylenes **1** and **2** on the viability and proliferative capacity of HGC-27 cells and tried to find suitable non-cytotoxic concentrations for subsequent experiments. The viability of HGC-27 cells was analyzed using an MTT assay, as shown in Figure 2A; despite being exposed to 6.25–25 µM of polyacetylenes **1** or **2** for 48 h, no significant difference (*p* > 0.05) in cell viability was observed compared with the control group. Furthermore, the proliferative potential of HGC-27 cells was evaluated by a colony formation assay, as shown in Figure 2B,C; the colony formation rate of HGC-27 cells following polyacetylene **1** or **2** treatment also had no substantial difference (*p* > 0.05) from that of the control group. These results indicated that 25 µM and below could be the non-cytotoxic concentrations of polyacetylenes **1** and **2** and were used in the following experiments.

### 2.3. Effect of Treatment on the Wound-Healing Capacity of HGC-27 Cells

A wound-healing assay is a simple and highly reproducible method to appraise the mass movement of cancer cells. In this study, the effects of polyacetylenes **1** and **2** on the migratory viability of HGC-27 cells in vitro were evaluated using a wound-healing assay. As shown in Figure 3, HGC-27 cells were treated with polyacetylene **1** or **2** at 6.25, 12.5, and 25 µM for 24 h. Compared with the original (0 h) wound width, both polyacetylenes **1** and **2** could significantly inhibit the motility of HGC-27 cells in a concentration-dependent manner, while polyacetylene **1** demonstrated a greater potential for inhibiting cell migration (*p* < 0.05).

### 2.4. Effect of Treatment on the Migration and Invasion of HGC-27 Cells

Cell migration and invasion are pivotal steps in tumor cells as they disseminate from the primary tumor, invade across the basement membranes and endothelial walls, and, finally, colonize distant organs [20]. A transwell assay is the most common method used to detect the migration and invasion of tumor cells. In a transwell migration assay, the migration ability of tumor cells is reflected by the attraction of FBS or certain chemokines to cells. However, in a transwell invasion assay, Matrigel simulates the extracellular matrix in vivo, and only cells that secrete matrix metalloproteinases to degrade the matrix glue can penetrate the lower chamber. Thus, transwell migration assays and Matrigel transwell invasion assays were conducted to analyze the inhibitory effects of polyacetylenes **1** and **2** on the migration and invasiveness of HGC-27 cells. The results demonstrated that polyacetylenes **1** and **2** could inhibit the migration and invasiveness of HGC-27 cells in a dose-dependent manner (Figure 4) (*p* < 0.05). Furthermore, the inhibitory effect of the trans structure polyacetylene **1** was even better.

### 2.5. Effect of Treatment on the Adhesion Ability of HGC-27 Cells

Collective cell migration requires cell–cell and cell–ECM adhesions. A cell adhesion assay was used for assessing the adhesion ability of the external environment of cells. Compared with the control, polyacetylenes **1** and **2** significantly decreased the number of adherent cells (*p* < 0.05), while the inhibition effect of polyacetylene **1** visibly exceeded that of polyacetylene **2** (Figure 5). Since polyacetylene **1** had a more prominent effect on inhibiting the migration, invasion, and adhesion of HGC-27 cells, it was selected for the subsequent mechanistic investigation.

### 2.6. Polyacetylene ***1*** Treatment Reversed the EMT in HGC-27 Cells

EMT was shown to be closely related to tumor initiation, invasion, metastasis, and resistance to therapy. EMT dynamics could mediate its biological impact on tumor metastasis by inhibiting the expression of epithelial markers (e.g., E-cadherin) and upregulating the expression of mesenchymal markers (e.g., Vimentin and Snail), thereby remolding the cytoskeleton and cell membrane, endowing cancer cells with increased motility and invasiveness, and causing the metastasis and spread of lesions [21]. Therefore, immunofluorescence assays and Western blot analysis were conducted in this study to examine the effect of polyacetylene **1** on EMT markers in HGC-27 cells. Immunofluorescence staining revealed that E-cadherin expression was increased and Vimentin expression was decreased in the HGC-27 cells treated with polyacetylene **1** (Figure 6). Simultaneously, the Western blot outcome further verified this point and displayed a significant decrease in Snail. These findings indicated that polyacetylene **1** inhibited the EMT process in HGC-27 cells.

### 2.7. Polyacetylene ***1*** Treatment Suppressed the Wnt/*β*-catenin Signaling Pathway in HGC-27 Cells

The Wnt/*β*-catenin signaling pathway plays a vital role in regulating the migration and invasion of tumor cells. The key molecule of the Wnt signaling pathway *β*-catenin is involved in cell adhesion through forming cadherin–catenin complexes, as well as in gene transcription through interacting with transcription factors. The nuclear accumulation of *β*-catenin, which is a hallmark of Wnt signaling activation, is found in more than 50% of GC [22,23]. Consequently, immunofluorescence assays and Western blot analysis were employed to detect the effect of polyacetylene **1** on the nuclear localization proportion of *β*-catenin and the expression of Wnt signaling pathway molecules. Compared with the control group, the expression of *β*-catenin, p-*β*-catenin, GSK3*β*, and p-GSK3*β* in the polyacetylene **1** treatment groups was obviously decreased in a dose-dependent manner (Figure 7A). The expressions of the *β*-catenin target genes c-Myc, ICAM-1, VCAM-1, cyclin D1, MMP-2, MMP-7, MMP-9, and MMP-14 were also significantly decreased (Figure 7A). In addition, the expression of nuclear *β*-catenin of the treatment group was significantly lower than that of the control group, which was also verified by the Western blot and immunofluorescence results (Figure 7B). The above results confirmed that polyacetylene **1** could inhibit the activation of *β*-catenin, promote its degradation, and block the Wnt/*β*-catenin signaling pathway in HGC-27 cells.

### 2.8. Polyacetylene ***1*** Treatment Suppressed the Hippo–YAP Signaling Pathway in HGC-27 Cells

The Hippo–YAP signaling pathway was found to be bound up with EMT and tumor metastasis in GC. Dephosphorylation of YAP/TAZ can induce the expression of target genes through nuclear translocation and cooperates with target factors to activate multiple oncogenes, promoting tumor progression, migration, anti-apoptosis, and metastasis [24,25,26,27]. The expression of LATS1 is downregulated and negatively associated with YAP in GC tissues, whereas the silencing of YAP reduces the growth and invasion in HGC-27 cells [28,29]. Accordingly, immunofluorescence assays and Western blot analysis were employed to detect the effect of polyacetylene **1** on the nuclear localization proportion of YAP/TAZ and the expression of Hippo–YAP signaling pathway molecules. The Western blotting analysis suggested that polyacetylene **1** significantly decreased the expression of MST1, p-MST1, YAP, p-YAP, LAST1, YAP/TAZ, MAPK, Cyr61, Cox2, and Sox2 while simultaneously increasing the expression of p-LATS1 (Figure 8A). In addition, the expression of nuclear YAP/TAZ of the treatment group was significantly lower than that of the control group, which was also confirmed by the Western blot and immunofluorescence results (Figure 8B). The above results confirmed that polyacetylene **1** can inhibit the activation of YAP/TAZ, promote its degradation, and block the Hippo/YAP signaling pathway in HGC-27 cells.

## 3. Discussion

Invasion and metastasis are the most defining features of gastric cancer malignancy and the leading causes of patient mortality. Blocking the acquisition of invasiveness and migration capability with small molecules represents a novel and promising treatment strategy for advanced gastric cancer. Natural-product-derived compounds have since long been recognized as important sources of anticancer drugs, some of which have been shown to exhibit promising anti-metastasis activities by suppressing key molecular features that uphold tumor cell aggressiveness. For example, matrine, which is a major compound isolated from *Sophora flavescens* Ait, exhibits strong anti-invasive activity against breast cancer cells [30]. Gigantol, which is a bibenzyl compound obtained from *Dendrobium draconis*, was demonstrated to suppress the migration and invasion of lung cancer cells [31].

Naturally occurring polyacetylenes are characterized by the presence of two or more carbon–carbon triple bonds in their carbonic skeleton and are considered important active ingredients in many medicinal and edible plants, such as *Bidens pilosa* L., *Atractylodes lancea* (Thunb.) DC., and *Carthamus tinctorius* L. Polyacetylene analogs have an alkyne-conjugated macrocyclic structure, which, upon activation, can form benzene-type free radicals that can easily enter the interior of tumor tissues, and upon targeting in tumor cells, rapidly seize the hydrogen atoms of DNA, cracking the DNA strands, and thus, producing remarkable anti-tumor activity. In addition, due to the carbon–carbon double bond and triple bond, polyacetylene compounds possess structural diversity and are easy-to-form isomers, which may exert different pharmacological effects. In the present study, we first revealed the excellent anti-metastasis potential of polyacetylenes **1** and **2** on human undifferentiated gastric cancer HGC-27 cells in vitro. Meanwhile, these two compounds also have other unique properties, such as low toxicity for normal gastric mucosal cells and kidney cells, small molecular weight for entering solid tumors expediently, and a succinct structure for easy chemical synthesis. With these characteristics, polyacetylenes **1** and **2** would be the high-potential candidate drugs for anti-gastric cancer metastasis.

As shown in Figure 9, the potential molecular mechanisms underlying the anti-migration and invasion effects of HGC-27 cells were deeply clarified by focusing on the process of EMT and related signaling pathways. The treatment of gastric cancer HGC-27 cells with non-toxic concentrations of polyacetylene **1** significantly suppressed EMT markers, namely, Vimentin and Snail, while it enhanced the level of E-cadherin, indicating that polyacetylene **1** could reverse the EMT process. At the advanced stage of tumorigenesis, the expression of EMT-related transcription factor Snail is upregulated, which, in turn, inhibits the expression of E-cadherin and increases the expression of Vimentin [32]. Changes in the expressions of these epithelial and mesenchymal markers result in the downgrade of cell polarity, reduce the adhesion between transitional cells and adjacent epithelial cells, and increase the secretion of enzymes that degrade the extracellular matrix, thereby facilitating the shedding of tumor cells from their primary site and stimulating metastasis [33].

Wnt/*β-*catenin signaling is considered a regulator of EMT. Hence, the effects of polyacetylene **1** on the Wnt/*β-*catenin signaling pathway were subsequently revealed. The results showed that polyacetylene **1** inhibited *β*-catenin entry into the nucleus through two pathways: the inhibition of GSK3*β* expression and phosphorylation, and the promotion of ubiquitinated degradation of *β*-catenin phosphorylation, which, in turn, inhibited downstream transcriptional expressions of c-Myc, ICAM-1, VCAM-1, MMPs, etc. Our results suggested that polyacetylene **1** not only inhibits the Wnt/*β*-catenin signaling pathway but also accelerates the ubiquitination and degradation of *β*-catenin, thereby inhibiting EMT and metastasis of HGC-27 cells. In the cytoplasm, *β*-catenin binds to intracellular peptides in the cytoplasmic domain of E-cadherin to form the E-Cad/Cat complex, which mediates cell junctions and maintains cell polarity, while the phosphorylation of *β*-catenin inhibits the adhesion function of E-cadherin, resulting in unstable intercellular adhesion [34]. Wnt/*β*-catenin signaling plays a critical role in EMT regulation by downregulating the expression of E-cadherin, which subsequently leads to the release and activation of *β*-catenin. Wnt signaling enhances the expression of Snail by inhibiting its phosphorylation, thus inducing EMT in cancer cells [35].

The Hippo–YAP pathway is also involved in the EMT process of gastric cancer cells. Several molecules were reported to promote EMT or enhance tumor metastasis via the Hippo–YAP pathway in gastric cancer [36]. The growth, migration, and invasion of gastric cancer cells can be reduced by silencing YAP, while the ectopic expression of YAP promotes these [37]. Simultaneously, the Hippo–YAP pathway is inseparably associated with Wnt/*β-*catenin. The overlaps between the biological processes controlled by YAP/TAZ and Wnt/*β*-catenin suggest that these factors do not act in isolation, but may interact with each other. Phosphorylated-*β*-catenin promotes TAZ degradation by acting as TAZ to b-TrCP presenters [12]. By controlling the stability of *β*-catenin, Wnt also induces TAZ stabilization and TAZ nuclear entry in a manner that is independent of Hippo signaling. YAP and TAZ are integral components of the *β*-catenin destruction complex that serves as a cytoplasmic sink for YAP/TAZ. In addition, YAP can induce the inactivation of GSK3*β*, which, in turn, stabilizes cytoplasmic *β*-catenin and promotes the nuclear translocation of *β*-catenin [38]. Phosphorylated YAP/TAZ binds to *β*-catenin, resulting in *β*-catenin retention in the cytoplasm, thereby inhibiting the transcriptional activity of *β*-catenin [39]. In this study, polyacetylene **1** prevented YAP/TAZ from entering the nucleus through multiple pathways, including the inhibition of MST1 phosphorylation, enhancement of LATS1 phosphorylation, and reduction of YAP expression. The reduction in YAP/TAZ in the nucleus leads to decreases in ICAM-1, VCAM-1, Cyr61, and Sox2. Then, the migration and invasion abilities of HGC-27 cells were significantly inhibited. This was the first report that demonstrated that polyacetylene **1** can regulate the Hippo–YAP signaling pathway.

In conclusion, polyacetylenes **1** and **2** exhibited anti-migration, invasion, and adhesion activity on HGC-27 cells at their nontoxic concentrations by reversing the EMT process and inhibiting Wnt/*β*-catenin and Hippo/YAP signaling pathways. Future research involving animal experiments is needed to comprehensively elucidate the stability, bioavailability, and metabolism of polyacetylenes **1** and **2**. These findings will be helpful for the future applications of polyacetylenes **1** and **2** as promising therapeutic agents for gastric cancer.

## 4. Materials and Methods

### 4.1. Cells and In Vitro Treatments

Human gastric carcinoma HGC-27 cells, human gastric carcinoma BGC-823 cells, human gastric carcinoma NCI-N87 cells, human breast cancer MDA-MB-231 cells, human hepatocellular carcinoma HepG2 cells, human colon cancer HCT-116 cells, human cervical cancer Ca Ski cells, human pancreatic cancer PANC-1 cell, human lung cancer A549 cells, human nasopharyngeal carcinoma CNE-2 cells, human gastric epithelial GES-1 cells, and Madin–Darby canine kidney MDCK cells were purchased from the cell bank of the Shanghai Institute for Biological Sciences, Chinese Academy of Sciences (Shanghai, China) and maintained in RPMI-1640 (Gibco BRL Life Technologies, Grand Island, NY, USA) or DMEM (Gibco BRL Life Technologies, Grand Island, NY, USA) culture medium supplemented with suitable fetal bovine serum (FBS, Zhejiang Tianhang Biological Technology Co., Ltd., Hangzhou, Zhejiang, China) at 37 °C with 5% CO_2_ in a humidified incubator.

### 4.2. Chemical and Reagents

The roots of *Bidens pilosa* L. were extracted three times in 95% ethanol for 7 d per extraction, then extracted with petroleum ether, ethyl acetate, and n-butanol in turn. The extraction solutions of different polarities were combined and concentrated under reduced pressure to obtain the petroleum ether extraction part, the ethyl acetate extraction part, and the n-butanol extraction part. The petroleum ether extraction site was separated into 17 sub-streams (Fr. A–Q) via column chromatography using a step-gradient eluent mixture of petroleum ether-ethyl acetate (from 100:0 to 0:100), of which Fr. A, Fr. B, Fr. F, Fr. H, Fr. I, Fr. K, Fr. N, and Fr. O had obvious characteristic absorption peaks of polyacetylene, and were the main peaks. The fragment Fr.F (357 mg) was purified using semi-preparative HPLC (acetonitrile:water = 55:45) to obtain the compound polyacetylene **2** (5.3 mg). Fr. H was purified using semi-preparative HPLC (acetonitrile:water = 55:45) to obtain the compound polyacetylene **1** (125.3 mg). To view the polyacetylenes **1** and **2** ^1^H and ^13^C NMR data, see the Appendix A.

Rabbit monoclonal antibodies against E-cadherin (24E10), Vimentin (D21H3), Snail (C15D3), *β*-catenin (D10A8), p-*β*-catenin (D2F1), GSK-3*β* (D5C5Z), p-GSK3*β* (D3A4), MST1 (D8B9Q), p-MST1 (E7U1D), LATS1 (C66B5), p-LATS1 (D57D3), YAP (D8H1X), p-YAP (D9W2I), YAP/TAZ (D24E4), MMP2 (D4M2N), MMP7 (D4H5), MMP9 (D6O3H), MMP14 (D1E4), c-Myc (D3N8F), ICAM-1 (E3Q9N), VCAM-1 (D8U5V), MAPK (137F5), p-MAPK (D13.14.4E), Sox2 (D6D9), Cox2 (D5H5), Cyr61 (D4H5D), GAPDH (D16H11), and Histone-H3 (D1H2) were used at a dilution of 1:1000. HRP-conjugated goat anti-rabbit IgG secondary antibody (98164) and FITC-conjugated goat anti-rabbit IgG secondary antibody (17A2) were used at dilutions of 1:5000 and 1:100, respectively. All of these were purchased from Cell Signaling Technology (Dancer, MA, USA). Matrigel was purchased from BD Biosciences (Franklin Lakes, NJ, USA).

### 4.3. Cytotoxicity Evaluation

The MTT method was used for the cytotoxicity evaluation of polyacetylenes **1** and **2**. Cells in the logarithmic growth phase were inoculated into the 96-well plate at 1 × 10^5^ cells/mL and treated with polyacetylene **1** or **2** at different concentrations for 48 h. Then, 20 µL 5 mg/mL MTT (BS0328; Amersco, Spokane, WA, USA) was added to each well and the cells were incubated for 4 h. Then, 150 µL DMSO (Sigma-Aldrich, St. Louis, MO, USA) was used to dissolve formazan. The absorbance of each well was measured using a microplate reader at 490 nm. Cytotoxicity was expressed as a 50% inhibition concentration (IC_50_).

### 4.4. Clone Formation Assay

HGC-27 cells in the logarithmic growth phase were inoculated into a 6-well plate at 400 cells per well and then incubated with polyacetylene **1** or **2** for 24 h. The medium was replaced with fresh medium every 2 days. After 14 days of incubation, the colonies were fixed in 4% paraformaldehyde and stained in 0.1% crystal violet solution. Images were captured, colonies consisting of more than 50 cells were counted, and data analysis was performed.

### 4.5. Wound-Healing Assay

HGC-27 cells in the logarithmic growth phase were inoculated into a 6-well plate at 5 × 10^5^ cells/mL. A 200 μL pipette tip was used to draw a vertical line in the direction perpendicular to the parallel lines when HGC-27 cells were adherent to 90%. The cells were treated with polyacetylene **1** or **2** for 24 h separately. Migration and cell movement throughout the wound area was observed with an inverted optical microscope (Olympus, Tokyo, Japan) and imaged using a camera attached to the microscope at 100× magnification at 6, 12, and 24 h. The percentage of wound closure was analyzed using ImageJ.

### 4.6. Cell Adhesion Matrix Capability Assay

HGC-27 cells were seeded into a 96-well plate at 1 × 10^5^ cells/mL and treated with polyacetylene **1** or **2** for 24 h separately. The 96-well plates were precoated with PBS-diluted Matrigel and blocked in advance. The cells were inoculated into the Matrigel-coated wells for 2 h. After incubation, the cells were fixed and stained in DAPI staining solution. The average number of adhered cells in five fields was counted using a fluorescence microscope (Olympus, Tokyo, Japan).

### 4.7. Transwell Migration/Invasion Assay

The HGC-27 cell line possessed a high degree of metastatic potential. Cell migration and invasion treated with polyacetylenes **1** and **2** were assessed by using the Matrigel invasion system. Transwell chambers (8 μm pore size, polycarbonate filters, 6.5 mm diameter; Corning Costar) in 24-well plates were coated with polymerized Matrigel (BD Biosciences, Franklin Lakes, NJ, USA). A total of 600 μL starved treated cells were placed in the upper chamber at 5 × 10^5^ cells/mL. Medium containing 20% FBS and polyacetylene **1** or **2** were added to the lower chamber as the chemoattractant. After incubation for 24 h, the invaded and migrated cells in the lower chamber were obtained via staining and counted.

### 4.8. Immunofluorescence Assays

Cells were inoculated on the chamber slides of 24-well culture plates and then treated with polyacetylene **1** for 24 h. After incubation, the cells were fixed, permeated, blocked, and incubated with a specific primary antibody and a secondary antibody. The nuclei were counter-stained with DAPI staining solution. After mounting with an anti-fluorescence quencher, images were captured using a fluorescence microscope (Olympus, Tokyo, Japan).

### 4.9. Western Blot Analysis

To determine the effect of polyacetylene **1** treatment on protein expression in HGC-27 cells, total protein was extracted after treating the cells with polyacetylene **1** for 24 h. Total protein was separated using SDS-PAGE gel electrophoresis and transferred to PVDF membranes according to their molecular weight. After blocking and incubating with a specific primary antibody and a secondary antibody, the signal detection was performed using ECL (Beyotime Biotechnology, Shanghai, China) and observed with a Tanon 5200 luminescence imaging system (Tanon Technology Co., Ltd., Xinjiang, China). Grayscale analysis was performed using Image J, and the experiments were performed three times in parallel.

### 4.10. Statistical Analysis

All the quantitative data were presented as mean ± SD and analyzed using ImageJ version 2.1 (National Institutes of Health) and SPSS standard version 20.0 software (SPSS, Inc., Chicago, IL, USA). The comparison of different experimental groups was performed using a t-test or one-way ANOVA analysis. *p* < 0.05 was considered statistically significant.

## Figures and Tables

**Figure 1 molecules-28-01837-f001:**
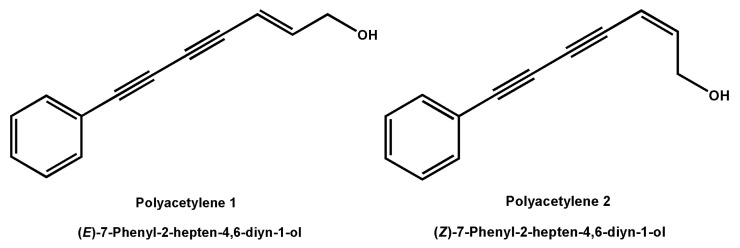
Structures of polyacetylenes **1** and **2**.

**Figure 2 molecules-28-01837-f002:**
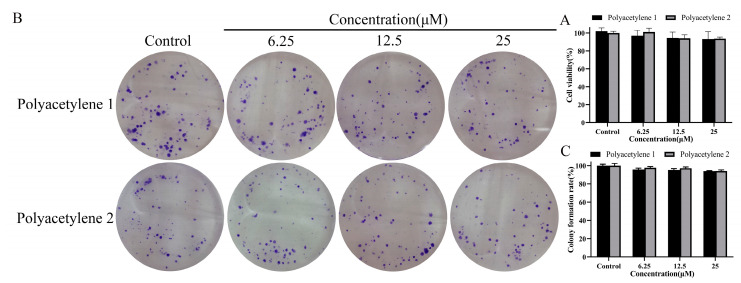
Determination of non-cytotoxic concentrations. (**A**) Cell viability of HGC-27 cells treated with 25, 12.5, and 6.25 µM polyacetylenes **1** and **2** for 48 h as determined using an MTT assay. (**B**) Representative stained colony plates of HGC-27 cells treated with polyacetylenes **1** and **2**. (**C**) Colony formation rate of HGC-27 cells treated with polyacetylenes **1** and **2**. Results were expressed as the mean ± SEM (n = 3). No significant difference compared with the control group.

**Figure 3 molecules-28-01837-f003:**
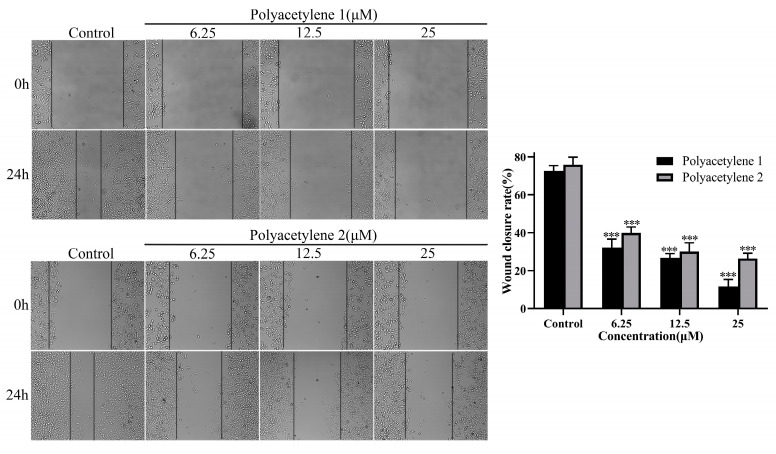
Effects of polyacetylenes **1** and **2** on the wound-healing capacity of HGC-27 cells. The HGC-27 cells were treated with 25, 12.5, and 6.25 µM polyacetylene **1** or **2** for 24 h (magnification: 100×). Compared with the control group, polyacetylenes **1** and **2** inhibited the wound-healing ability of HGC-27 in a dose-dependent manner, in which the inhibition ability of polyacetylene **1** was more significant. Significant differences compared with the control group are designated as *** *p* < 0.001. Results are expressed as the mean ± SEM (n = 3).

**Figure 4 molecules-28-01837-f004:**
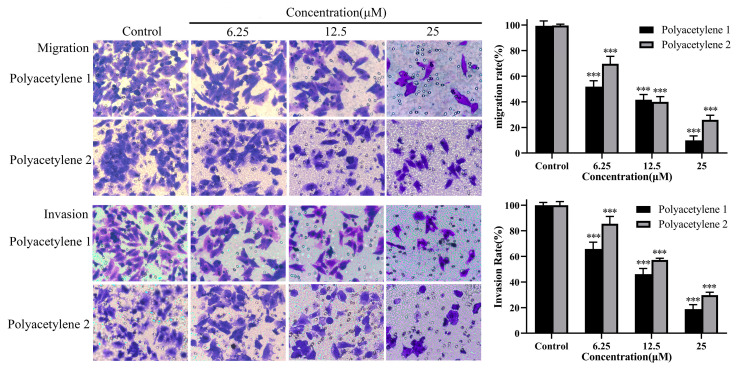
Effects of polyacetylenes **1** and **2** on the migration and invasion capabilities of HGC-27 cells. The HGC-27 cells were treated with 25, 12.5, and 6.25 µM polyacetylenes **1** and **2** for 24 h (magnification: 400×). Compared with the control group, polyacetylenes **1** and **2** inhibited the wound-healing ability of HGC-27 in a dose-dependent manner, in which the inhibition ability of polyacetylene **1** was more significant. Significant differences compared with the control group are designated as *** *p* < 0.001. Results are expressed as the mean ± SEM (n = 3).

**Figure 5 molecules-28-01837-f005:**
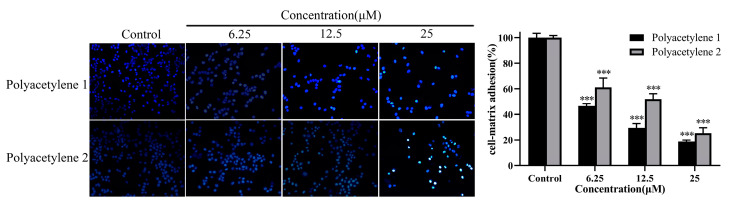
Effects of polyacetylenes **1** and **2** on the cell adhesion matrix capability of HGC-27 cells. Cells were treated with 25, 12.5, and 6.25 µM of polyacetylenes **1** and **2** for 24 h. The results are shown at 100× magnification. Compared with the control group, polyacetylenes **1** and **2** inhibited the wound-healing ability of HGC-27 in a dose-dependent manner, in which the inhibition ability of polyacetylene **1** was more significant. Significant differences compared with the control group are designated as *** *p* < 0.001. Results are expressed as the mean ± SEM (n = 3).

**Figure 6 molecules-28-01837-f006:**
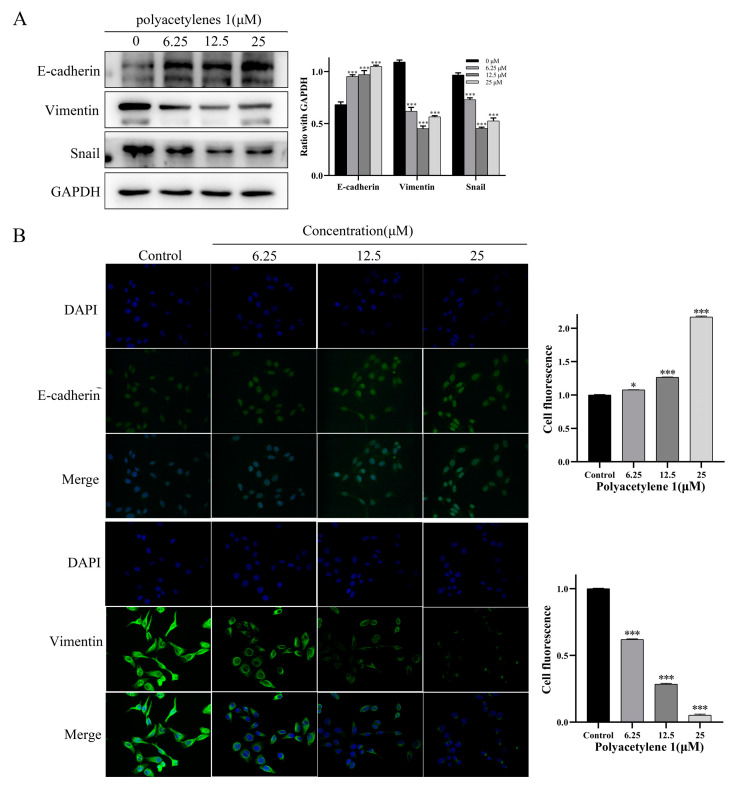
Effects of polyacetylene **1** on the EMT marker molecule protein in HGC-27 cells. (**A**) Effects on the expression of E-cadherin, Vimentin, and Snail in HGC-27 cells treated with 25, 12.5, and 6.25 µM of polyacetylene **1** for 24 h. E-cadherin, Vimentin, and Snail were measured using the Western blotting method. GAPDH was used to ensure an equal amount of loaded protein. (**B**) Immunofluorescence images of E-cadherin and Vimentin in HGC-27 cells treated with 25, 12.5, and 6.25 µM of polyacetylene **1** for 24 h. Nuclear counterstaining with DAPI is shown separately. Scale bars: 200 μm. Significant differences compared with the control group are designated as * *p* < 0.05 and *** *p* < 0.001. Results are expressed as the mean ± SEM (n = 3).

**Figure 7 molecules-28-01837-f007:**
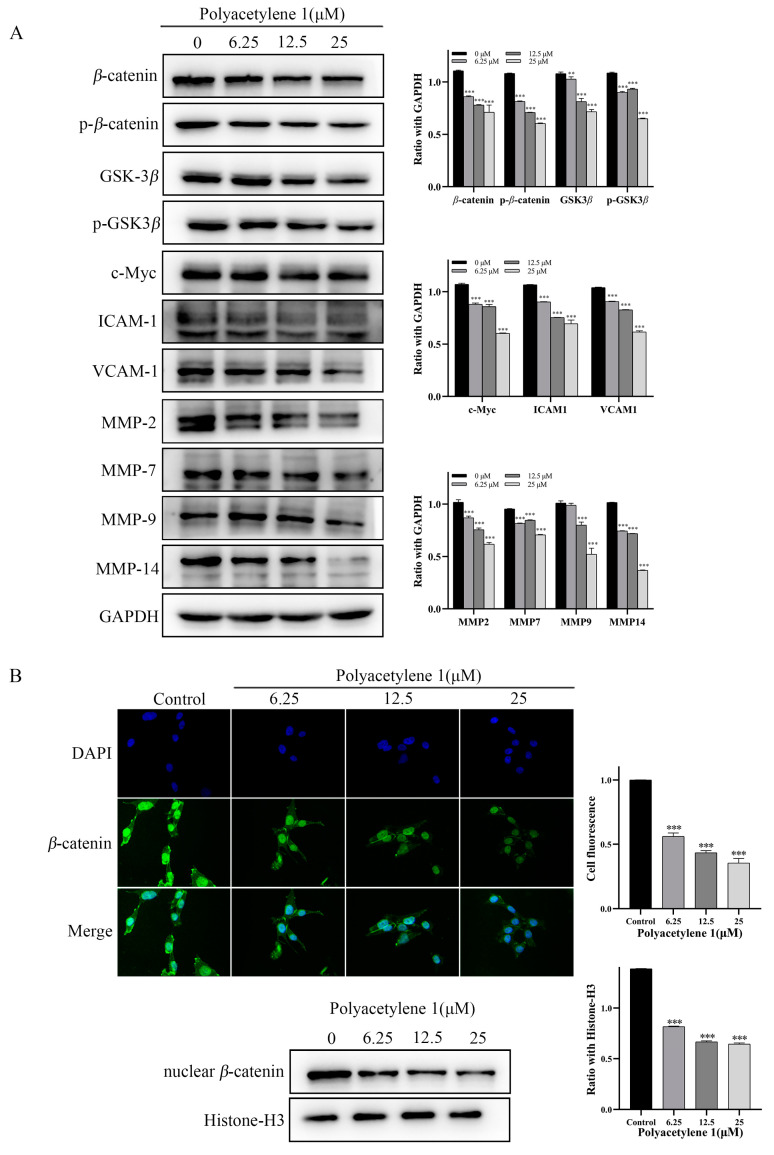
Effects of polyacetylene **1** on the Wnt/*β*-catenin signaling pathway in HGC-27 cells. (**A**) Western blot results of Wnt/*β*-catenin signaling pathway-related proteins in total lysates of HGC-27 cells treated with 25, 12.5, and 6.25 µM of polyacetylene **1** for 24 h. GAPDH was used to ensure an equal amount of loaded protein. (**B**) Immunofluorescence images and Western blot results of nuclear *β*-catenin in HGC-27 cells treated with 25, 12.5, and 6.25 µM of polyacetylene **1** for 24 h. Nuclear counterstaining with DAPI was shown separately. Scale bars: 200 μm. Histone-H3 was used to ensure an equal amount of loaded protein. Significant differences compared with the control group are designated as ** *p* < 0.01 and *** *p* < 0.001. Results are expressed as the mean ± SEM (n = 3).

**Figure 8 molecules-28-01837-f008:**
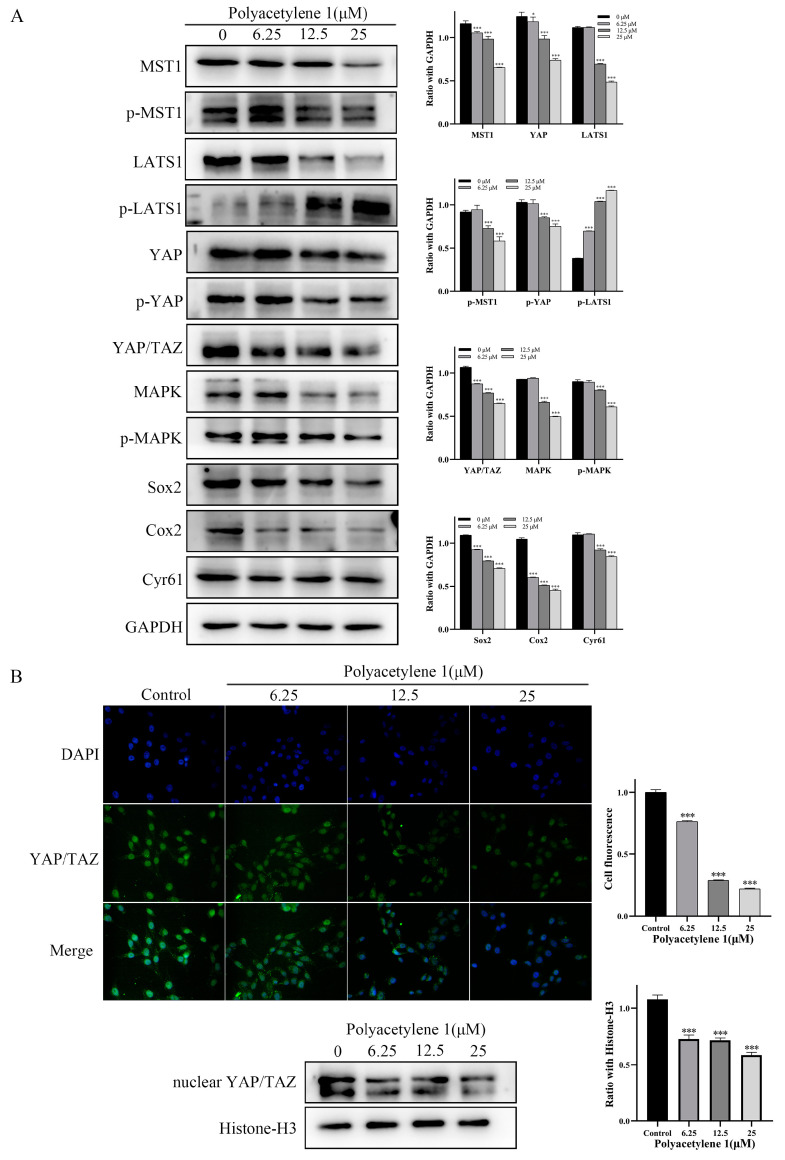
Effects of polyacetylene **1** on the Hippo/YAP signaling pathway in HGC-27 cells. (**A**) Western blot results of Hippo/YAP-signaling-pathway-related proteins in total lysates of HGC-27 cells treated with 25, 12.5, and 6.25 µM of polyacetylene **1** for 24 h. GAPDH was used to ensure an equal amount of loaded protein. (**B**) Immunofluorescence images and Western blot results of nuclear YAP/TAZ in HGC-27 cells treated with 25, 12.5, and 6.25 µM of polyacetylene **1** for 24 h. Nuclear counterstaining with DAPI is shown separately. Scale bars: 200 μm. Histone-H3 was used to ensure an equal amount of loaded protein. Significant differences compared with the control group are designated as * *p* < 0.05 and *** *p* < 0.001. Results are expressed as the mean ± SEM (n = 3).

**Figure 9 molecules-28-01837-f009:**
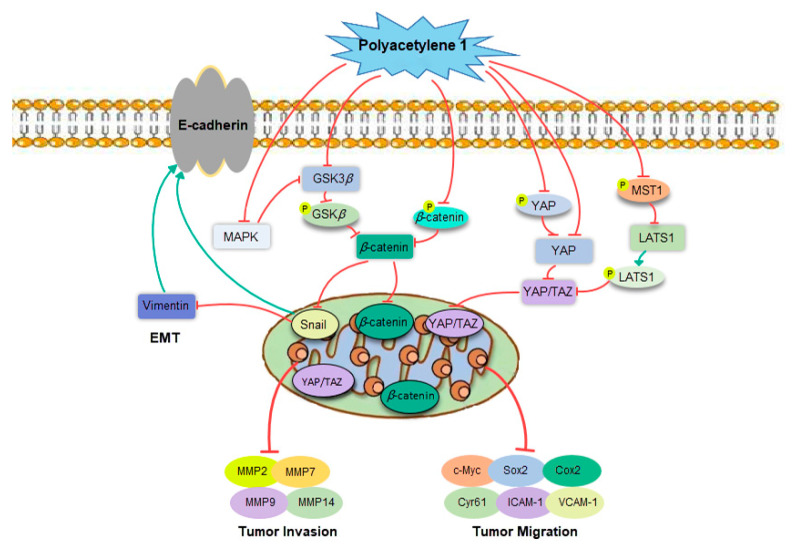
Polyacetylene **1** regulated tumor invasion and migration via various signaling pathways.

**Table 1 molecules-28-01837-t001:** The cytotoxic activity of polyacetylene **1** against tumor cell lines and normal cell lines.

Cell Line	HGC-27	MDA-MB-231	HepG2	BGC-823	HCT-116	Ca Ski
IC_50_ (μM)	52.83	81.88	108.7	126.5	164.1	166.6
Cell line	PANC-1	A549	NCI-N87	AML12	GES-1	MDCK
IC_50_ (μM)	171.4	>200	>200	73.92	160.6	171.9

Polyacetylene **1** displayed relatively obvious proliferation inhibitory activity against HGC-27 cells, with an IC_50_ value of 52.83 μM, while polyacetylene **2** was inactive in all tested cells, with IC_50_ values greater than 200 μM.

## Data Availability

The datasets used and analyzed during the current study are available from the corresponding author upon reasonable request.

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
