# Peer review of "Polyacetylene Isomers Isolated from Bidens pilosa L. Suppress the Metastasis of Gastric Cancer Cells by Inhibiting Wnt/β-Catenin and Hippo/YAP Signaling Pathways"

_molecules, 2023, doi:10.3390/molecules28041837_

Round 1
Reviewer 1 Report
In this paper, the authors demonstrate that polyacetylene 1 and 2 from Bidens pilosa L. exhibit specific antimetastatic properties in human HGC-27 gastric cancer cells. Overall, the paper is potentially interesting. However, I would like to raise the following concerns:
1. Lane 61. The authors must provide a reference about the characterization of plant extract and the extraction methods for two compounds. Alternatively, these details should be clearly explained in the materials and methods section.
2. Lane 104 and Figure 2. Panels of Figure 2 should be arranged according to the order of citation in the text.
3. The bar graphs of Figures 7 and 8 are difficult to read. Please increase the readability of the graphics.
4. A final figure or scheme summarizing the effects of the two molecules on the various identified pathways involved in the metastatic processes would improve understanding of the results.
5. Paragraph 4.1. Please specify the culture medium used for each cell lines.
6. Paragraph 4.2. Please provide the product (catalogue) number for each antibody.
7. Paragraph 4.3. The MTT assay should be better described. What is the wavelength used? What instrument was used for the test?
8. Paragraph 4.4. Please indicate the initial number of cells seeded for each well.
Reviewer 2 Report
Polyacetylenes isomers isolated from Bidens pilosa L. suppress 2 the metastasis of gastric cancer cells via inhibiting Wnt/β- 3 catenin and Hippo/YAP signaling pathways
In this study to explore the excellent anti-metastasis potential of these two polyacetylenes on human gastric cancer HGC-27 cells and the distinctive molecular mechanisms underlying their activities. Polyacetylene exhibited its anti-metastasis activities on HGC-27 cells through the reverse of EMT process and the suppression of the Wnt/β-catenin and Hippo/YAP signaling pathways. Overall, this manuscript is well designed and studied. Some suggestions for this paper are as follows:
Detailed comments to the author
1. The limitations of the work and its interpretation are not discussed.2. The current study needs to compare and discuss with their study.
Round 2
Reviewer 1 Report
The paper is now suitable for publication.